# Photonic chip-based low-noise microwave oscillator

Igor Kudelin[1,2 ✉], William Groman[1,2], Qing-Xin Ji[3], Joel Guo[4], Megan L. Kelleher[1,2], Dahyeon Lee[1,2], Takuma Nakamura[1,2], Charles A. McLemore[1,2], Pedram Shirmohammadi[5], Samin Hanifi[5], Haotian Cheng[6], Naijun Jin[6], Lue Wu[3], Samuel Halladay[6], Yizhi Luo[6], Zhaowei Dai[6], Warren Jin[4], Junwu Bai[5], Yifan Liu[1,2], Wei Zhang[7], Chao Xiang[4], Lin Chang[4], Vladimir Iltchenko[7], Owen Miller[6], Andrey Matsko[7], Steven M. Bowers[5], Peter T. Rakich[6], Joe C. Campbell[5], John E. Bowers[4], Kerry J. Vahala[3], Franklyn Quinlan[1,8] & Scott A. Diddams[1,2,8 ✉]

Numerous modern technologies are reliant on the low-phase noise and exquisite timing stability of microwave signals. Substantial progress has been made in the field of microwave photonics, whereby low-noise microwave signals are generated by the down-conversion of ultrastable optical references using a frequency comb[1–3]. Such systems, however, are constructed with bulk or fibre optics and are difficult to further reduce in size and power consumption. In this work we address this challenge by leveraging advances in integrated photonics to demonstrate low-noise microwave generation via two-point optical frequency division[4,5]. Narrow-linewidth self-injection-locked integrated lasers[6,7] are stabilized to a miniature Fabry–Pérot cavity[8], and the frequency gap between the lasers is divided with an efficient dark soliton frequency comb[9]. The stabilized output of the microcomb is photodetected to produce a microwave signal at 20 GHz with phase noise of −96 dBc Hz$^{-1}$ at 100 Hz offset frequency that decreases to −135 dBc Hz$^{-1}$ at 10 kHz offset—values that are unprecedented for an integrated photonic system. All photonic components can be heterogeneously integrated on a single chip, providing a significant advance for the application of photonics to high-precision navigation, communication and timing systems.

Low-noise microwave signals with high timing stability are a critical enabler of modern science and multiple technologies of broad societal impact. Positioning and navigation, advanced communications, high-fidelity radar and sensing, and high-performance atomic clocks are all dependent upon low-phase-noise microwave signals. These rapidly developing technologies are constantly intensifying the demand for microwave sources beyond current capabilities while imposing harsher restrictions on system size, weight, and power consumption (SWaP). In this landscape, photonic lightwave systems provide unique advantages over more conventional electronic approaches for generating low-noise microwaves. In particular, the extremely low-loss and high-quality factors of photonic resonators are fundamental to electromagnetic oscillators with the lowest noise and highest spectral purity[10]. Coupled to this is the introduction and rapid development of frequency combs in the last few decades that enable seamless coherent synthesis across the full electromagnetic spectrum[11]. This includes the frequency division of a 200–500 THz optical carrier down to a 10 GHz microwave with unrivalled long- and short-term stability[1–3,12].

However, a significant challenge of these approaches is the relatively large size and power consumption that restrict their use to laboratory environments. Greater impact and widespread use can be realized with a low-noise microwave generator that has a compact and portable form factor for operation in remote and mobile platforms. Our work addresses and overcomes this challenge through the optimal implementation of two-point optical frequency division (2P-OFD) with integrated photonic components as illustrated in Fig. 1. We provide a means to significantly reduce microwave phase noise in a volume of tens of millilitres instead of tens of litres while similarly reducing the required power by a factor 10$^3$ to the 1 W level.

All optical frequency division (OFD) systems start with a stable optical frequency reference. Typically, this is a laboratory fibre or solid-state laser that is frequency stabilized to a large evacuated Fabry–Pérot (F-P) cavity[10,13]. Instead, we introduce an optimal combination of low-noise chip-integrated semiconductor lasers[14,15] and a new F-P concept that can be miniaturized to less than 1 cm$^3$ and chip-integrated without the need for high-vacuum enclosure[8,16–18]. The frequency noise of two semiconductor lasers near 1,560 nm is reduced by 40 dB through self-injection

[1]National Institute of Standards and Technology, Boulder, CO, USA. [2]Department of Physics, University of Colorado Boulder, Boulder, CO, USA. [3]T. J. Watson Laboratory of Applied Physics, California Institute of Technology, Pasadena, CA, USA. [4]Department of Electrical and Computer Engineering, University of California, Santa Barbara, Santa Barbara, CA, USA. [5]Department of Electrical and Computer Engineering, University of Virginia, Charlottesville, VA, USA. [6]Department of Applied Physics, Yale University, New Haven, CT, USA. [7]Jet Propulsion Laboratory, California Institute of Technology, Pasadena, CA, USA. [8]Electrical Computer & Energy Engineering, University of Colorado Boulder, Boulder, CO, USA. ✉e-mail: igor.kudelin@colorado.edu; scott.diddams@colorado.edu

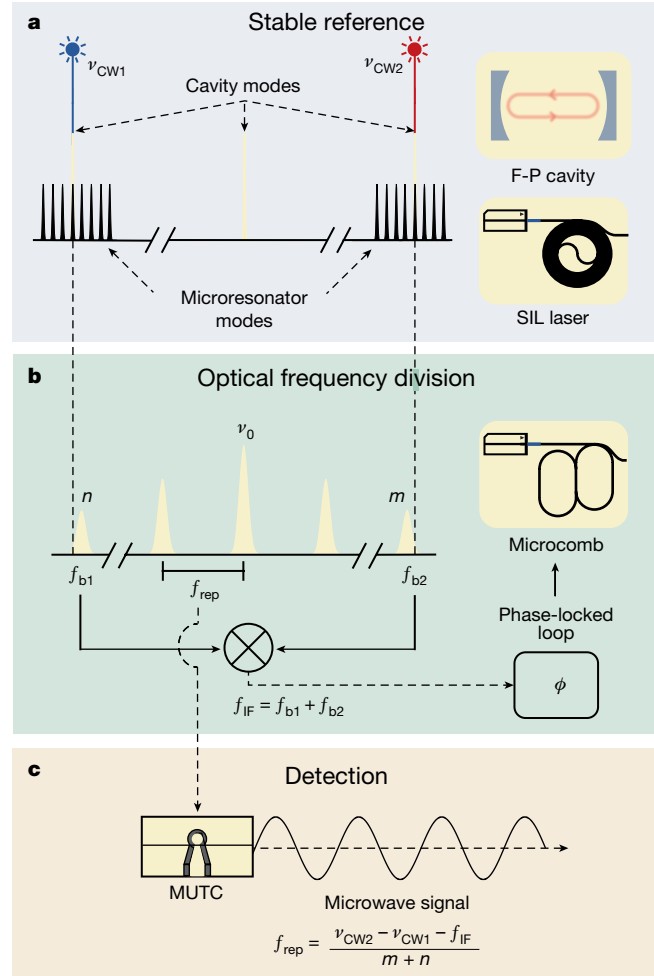

**a** Stable reference

$\nu_{CW1}$

Cavity modes

$\nu_{CW2}$

F-P cavity

SIL laser

Microresonator modes

**b** Optical frequency division

$\nu_0$

$n$ $m$

$f_{b1}$ $f_{rep}$ $f_{b2}$

$f_{IF} = f_{b1} + f_{b2}$

$\phi$

Phase-locked loop

Microcomb

**c** Detection

MUTC

Microwave signal

$$f_{rep} = \frac{\nu_{CW2} - \nu_{CW1} - f_{IF}}{m + n}$$

**Fig. 1 | Concept of 2P-OFD for low-noise microwave generation. a**, Two semiconductor lasers are injection locked to chip-based spiral resonators. The optical modes of the spiral resonators are aligned, using temperature control, to the modes of the high-finesse F-P cavity for PDH locking. **b**, A microcomb is generated in a coupled dual-ring resonator and is heterodyned with the two stabilized lasers. The beat notes are mixed to produce an intermediate frequency, $f_{IF}$, that is phase-locked by feedback to the current supply of the microcomb seed laser. **c**, An MUTC photodetector chip is used to convert the microcomb's optical output to a 20 GHz microwave signal.

locking (SIL) to high-Q (quality factor) $Si_3N_4$ spiral resonators[6,7]. This passive stabilization of the SIL laser enables further noise reduction, by up to 60 dB, through Pound–Drever–Hall (PDH) locking to a miniature F-P cavity, reaching the cavity's thermal noise limit[6,19].

In OFD, the optical phase noise of the reference is then reduced by the square of the ratio of its frequency to that of the microwave output. This is a powerful means for noise reduction by a factor as large as $(2 \times 10^{14}/1 \times 10^{10})^2 = 4 \times 10^8$ or equivalently, 86 dB. However, significant electrical power is required for generating a frequency comb spanning approximately 200 THz with mode spacing of 10–20 GHz. Instead, we use the simplified approach of 2P-OFD[4,5,20,21], with comb bandwidth on the order of 1 THz. This results in a lower division factor, but it also significantly reduced size and power requirements. Such a trade-off still allows us to reach an unprecedented microwave phase noise level with integrated photonics because of the intrinsic low noise of the optical references used.

In our system, the frequency division is implemented with another injection-locked laser that generates a microcomb in a zero group velocity dispersion (GVD) resonator, engineered by two coupled rings

in a Vernier configuration[9]. The microcomb operates without the need for optical amplification, and approximately 30% of the input pump power of 100 mW is efficiently transferred to the comb, which spans nearly 10 nm. 2P-OFD is then implemented by heterodyning the two SIL lasers with the closest comb teeth to produce two beat notes. These are mixed to provide a servo control signal at an intermediate frequency that is independent of the microcomb centre frequency. Upon phase locking of the intermediate frequency, the noise of the microcomb is dramatically reduced. Photodetection of the stabilized microcomb output with a high-power and high-linearity modified unitravelling carrier (MUTC) photodetector[22] provides a 20 GHz microwave signal with phase noise of −135 dBc Hz$^{-1}$ at 10 kHz offset frequency. This level of noise has not been achieved previously for a system that uses integrated photonic components. We note that the critical photonic devices used in our system can be further integrated to a single chip without the need for fibre or semiconductor amplifiers or optical isolators, providing ultrastable microwave generation in a compact form factor. This advance is important for future applications of high-performance microwave sources with compact size and low-power usage that will operate beyond research laboratories.

## Experiment and results

A technical illustration of the setup used for frequency comb stabilization and stable microwave generation is shown in Fig. 2. Here, we elaborate on the operation and characteristics of the key components, concluding with a description of how they function together cohesively to produce low-phase noise microwave signals.

### Miniature F-P cavity

The phase and frequency stability of the generated microwave signal is ultimately derived from that of the ultrastable optical reference. The lowest-noise optical references are lasers locked to vacuum-gap F-P cavities, where fractional frequency stability as low as $4 \times 10^{-17}$ has been demonstrated with 212-mm-long cryogenic cavity systems[10]. Instead, we use an integrable cavity design based on a compact, rigidly held cylindrical F-P optical reference cavity that supports fractional frequency stability at the $10^{-14}$ level[8]. Ultralow expansion glass with 1 m radius of curvature and an ultralow expansion glass spacer compose the 6.3-mm-long cavity with finesse of approximately 900,000 (Q ≈ 5 billion) and overall volume of less than 9 cm$^3$. The cavity is thermal noise limited for offset frequencies ranging from 1 Hz to 10 kHz. Moreover, the relative phase noise between two reference lasers locked to the same cavity takes advantage of large common-mode rejection (CMR), reaching 40 dB rejection for cavity modes spaced by 1 THz (ref. 18). When combined with 2P-OFD, the cavity noise is expected to be reduced by approximately 80 dB when projected onto the microwave carrier.

### Self-injection-locked lasers

To achieve high-stability performance, it is crucial to use narrow-linewidth and frequency-stable lasers. This is because electrical noise from the individual laser locking circuits does not experience CMR, and this noise is reduced only by the 2P-OFD. To address this issue and reach the thermal noise floor of the F-P cavity, we use SIL lasers that are both integrable and have noise performance equivalent to much larger laboratory fibre lasers[6,7,14]. With reference to Fig. 2, two commercial semiconductor distributed feedbacks (DFBs) are prestabilized by SIL to a high-Q $Si_3N_4$ spiral resonator[7]. When the forward and backward fields between the DFB lasers and the $Si_3N_4$ resonators are in phase, resonant backscattered light is fed back to the DFB, anchoring each laser wavelength to the corresponding $Si_3N_4$ resonance and significantly suppressing its frequency noise. This passive prestabilization of the DFB lasers is crucial to reach the thermal noise limit of the F-P cavity and reduce the high-frequency noise. The length and Q of the integrated resonator determine the ultimate phase noise of the

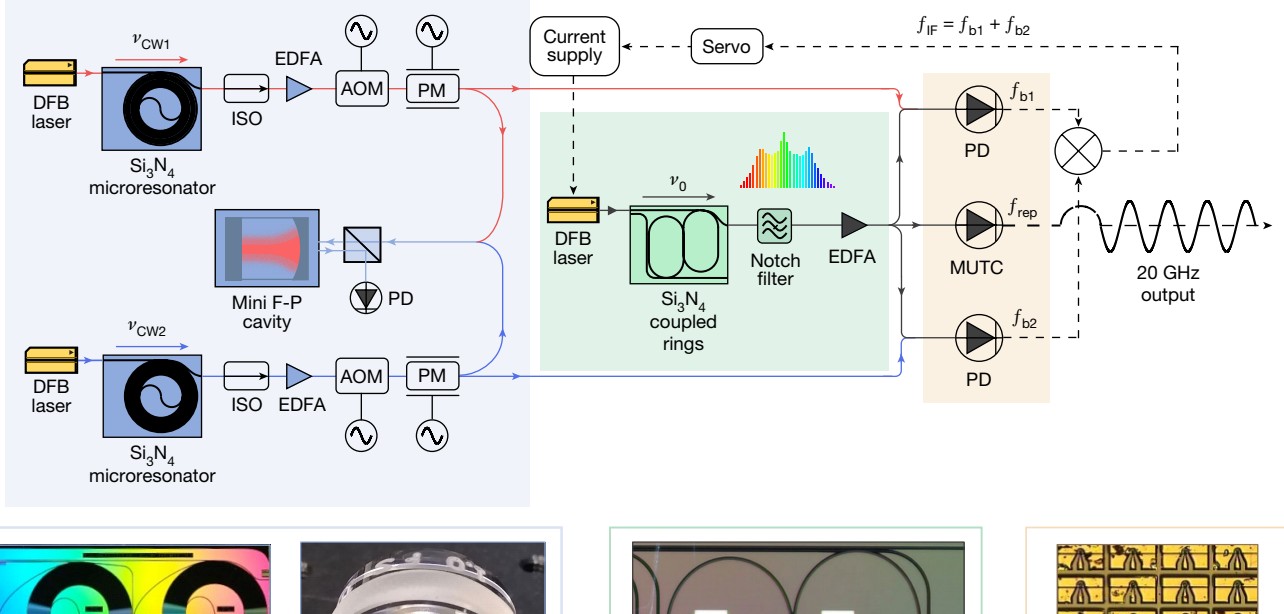

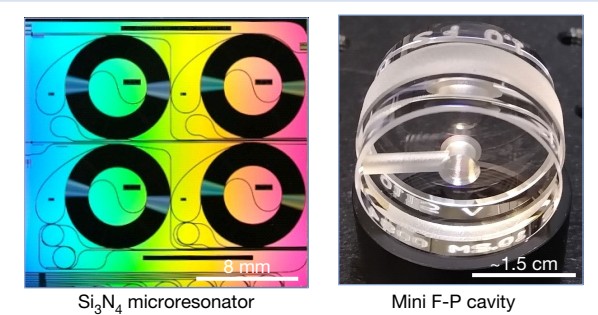

Si₃N₄ microresonator          Mini F-P cavity

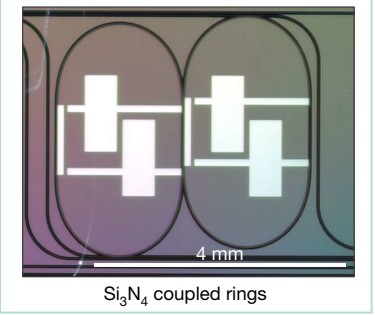

Si₃N₄ coupled rings

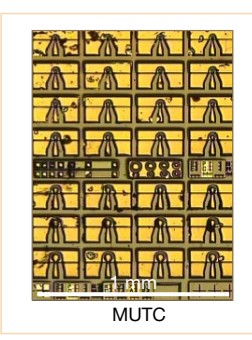

MUTC

**Fig. 2 | Experimental setup.** Two DFB lasers at 1,557.3 and 1,562.5 nm are self-injection-locked to Si₃N₄ spiral resonators, amplified and locked to the same miniature F-P cavity. A 6 nm broad-frequency comb with an approximately 20 GHz repetition rate is generated in a coupled-ring resonator. The microcomb is seeded by an integrated DFB laser, which is self-injection-locked to the coupled-ring microresonator. The frequency comb passes through a notch filter to suppress the central line and is then amplified to 60 mW total optical power. The frequency comb is split to beat with each of the PDH-locked SIL continuous wave references. Two beat notes are amplified, filtered and then mixed together to produce $f_{IF}$, which is phase-locked to a reference frequency. The feedback for microcomb stabilization is provided to the current supply of the microcomb seed laser. Lastly, part of the generated microcomb is detected in an MUTC detector to extract the low-noise 20 GHz signal. Photographs of the key photonic components used in low-noise microwave generation are in the lower panels. Scale bars (from left to right), 8 mm; approximately 1.5 cm; 4 mm; 1 mm. ISO, optical isolator; PM, phase modulator; PD, photodetector.

SIL laser[23], and here, we use a spiral with a length of 1.41 m that is fabricated on approximately 1 cm² of silicon. The intrinsic and loaded Q factors of the two spiral resonators are 164 and 126 million[7]. The outputs of the SIL lasers are amplified using commercial erbium-doped fibre amplifiers (EDFAs) to approximately 30 mW and then further stabilized via PDH locking to the miniature F-P cavity. In this setup, the primary actuator for PDH stabilization is an acousto-optic modulator; however, the PDH error signal is also fed back to the electro-optical modulator (EOM) to further increase the bandwidth and noise reduction of the PDH servo[24]. The in-loop phase noise of the PDH locking of SIL lasers is presented in Methods.

## Microcomb

Robust and low-noise optical frequency comb generation with 10–20 GHz repetition rate and broad optical coverage is challenging. Here, we use an Si₃N₄ microresonator fabricated at a complementary metal-oxide semiconductor (CMOS) foundry to generate mode-locked microcombs[14]. To produce dark soliton microcombs with higher bandwidth, we use a dual coupled-ring resonator with free-spectral range (FSR) of 20 GHz (ref. 9), where the zero GVD wavelength is tuned to approximately 1,560 nm using integrated heaters[25]. In addition, such microcomb states have high pump-to-comb conversion efficiency, benefiting microwave generation in a low-SWaP system. To generate the comb, a commercial semiconductor DFB laser without optical amplification is self-injection-locked to the dual coupled-ring resonator, which

narrows the linewidth of the pump laser and generates a reasonably stable 20 GHz comb[14,26,27]. Following the coupled rings, a notch filter is used to suppress the central (seed) comb line to avoid saturation in an EDFA, which amplifies the frequency comb up to 60 mW (Fig. 2). Note that the use of a drop port on the coupled-ring resonator can replace the notch filter. A typical comb spectrum after the EDFA is shown in Fig. 3b.

## Microcomb stabilization and microwave generation

As outlined above, in this work we use two-point locking to realize OFD for phase noise reduction. With appropriate fibre optic couplers and filters, we design a receiver system to separate heterodyne beat notes and 20 GHz microwave generation. The two beat notes between the microcomb and each continuous wave (CW) laser are given by $f_{b1} = (\nu_0 - n\,f_{rep}) - \nu_{CW1}$ and $f_{b2} = \nu_{CW2} - (\nu_0 + m\,f_{rep})$ (Fig. 1). Here, the comb modes are indexed with signed integers from the central (seed) frequency $\nu_0$. These beats are filtered, amplified and mixed together to produce the intermediate frequency ($f_{IF}$), which is then phase-locked to a stable microwave reference via feedback to the current of the microcomb seed laser[9]. The stabilization of $f_{IF}$ is the final step to generate a low-noise microwave via 2P-OFD: $f_{IF} = f_{b1} + f_{b2} = \nu_{CW2} - \nu_{CW1} - (n+m)f_{rep}$, where $n + m$ is the value of the OFD (32 in our case) that amounts to 30 dB of noise reduction. Note that two-point locking does not depend on the noise of the microcomb central frequency $\nu_0$. Thus, the instability of the microcomb repetition rate can be represented as

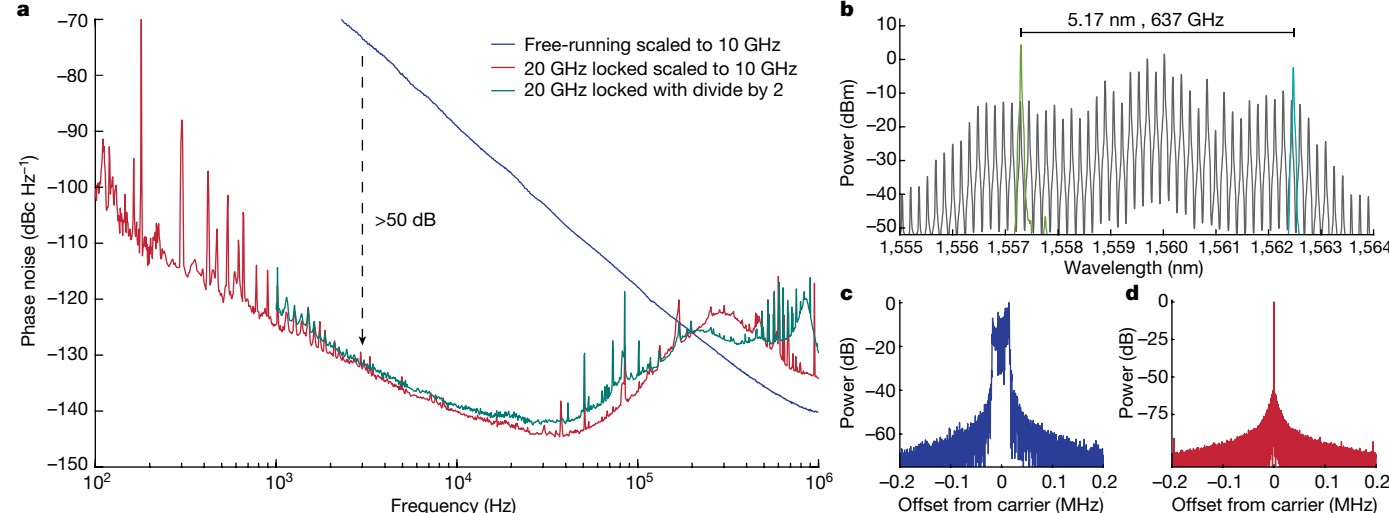

**Fig. 3 | Microcomb characterization. a**, Single side-band phase noise scaled to 10 GHz of free-running 20 GHz microcomb (blue), locked 20 GHz microwave (red) and locked 20 GHz microwave after regenerative frequency division by two (green). **b**, Optical spectrum of microcomb (grey) and SIL lasers (green and turquoise). **c,d**, Radio frequency spectra of 20 GHz signal free running (resolution bandwidth (RBW) 100 Hz; **c**) and locked (RBW 1 Hz; **d**).

$$\delta f_{\text{rep}}^2 = \frac{\delta(\nu_{\text{CW2}} - \nu_{\text{CW1}})^2 + \delta f_{\text{IF}}^2}{(n+m)^2}.$$

Although the phase noise of $f_{\text{IF}}$, $\nu_{\text{CW1}}$ and $\nu_{\text{CW2}}$ is reduced by 2P-OFD, their servo control and residual noise can be limiting factors in the achievable microwave phase noise (Methods).

The stabilized microcomb output is directed to an MUTC photodiode, which provides exceptional linearity and large microwave powers[28,29]. We tune the bias voltage of the MUTC operation for approximately 40 dB rejection of amplitude-to-phase noise conversion while generating a 20 GHz power of −10 dBm at 5 mA of average photocurrent. The 20 GHz microwave signal is filtered and amplified to +3 dBm, and it is sent to a measurement system, with results presented in Fig. 3a. Here, we have scaled the measured 20 GHz phase noise to 10 GHz by subtracting 6 dB. This yields −102 dBc Hz$^{-1}$ at 100 Hz, which decreases to −141 dBc Hz$^{-1}$ at 10 kHz. We also compare with a 10 GHz carrier that we generate from 20 GHz with a regenerative divide-by-two circuit. Compared with the free-running microcomb generation, we achieved more than 50 dB phase noise improvement for offset frequencies below 10 kHz. Additional details on the phase noise measurement and divider are in Methods.

## Discussion and further integration

Figure 4 places the level of phase noise we achieve in context with other photonic approaches, including recent works based on microcombs and mode-locked laser frequency combs. The comparison is classified by level of photonic integration of the microcomb source and pumping/reference lasers, as applicable. It is also noted that some of the microcomb systems require the assistance of a fibre-based frequency comb (Fig. 4ix,x)[30,31]. The phase noise performance of other systems, which could be chip integrated (Fig. 4ii,iii)[14,32], is more than 30 dB greater than the results we present, with the exception of the recent work by Sun et al. (Fig. 4vi)[33]. Other notable works on low-noise microwave generation in low-SWaP systems, which are not shown in Fig. 4, include 'quite point' operation[31,34–36], single-laser OFD[37] and high-end commercial products[38–40].

To the best of our knowledge, this work provides the best phase noise performance in the frequency range of 200 Hz to 40 kHz for microcomb-based systems. Importantly, it does so with integrated

photonic components that can all be further combined onto a single chip with total volume of the photonic components of approximately 1 cm$^3$. A concept of such a fully integrated system is shown in Fig. 5a and would consist of heterogeneously integrated lasers near

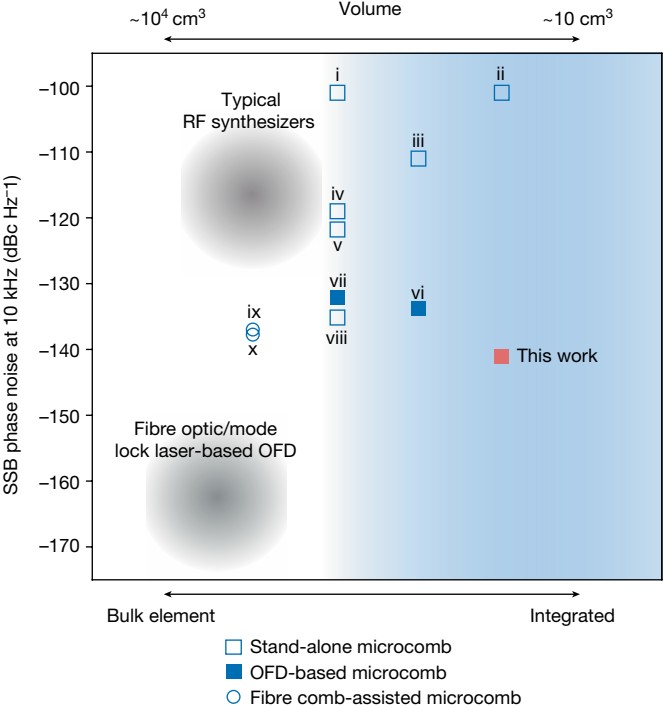

**Fig. 4 | Phase noise comparison of microwave generation based on microcombs.** The platforms are all scaled to 10 GHz carrier and categorized based on the integration capability of the microcomb generator and the reference laser source, excluding the interconnecting optical/electrical parts. Filled (blank) squares are based on the OFD (stand-alone microcomb) approach: (i) 22 GHz silica microcomb[50]; (ii) 5 GHz Si$_3$N$_4$ microcomb[14]; (iii) 10.8 GHz Si$_3$N$_4$ microcomb[32]; (iv) 22 GHz microcomb[51]; (v) MgF$_2$ microcomb[52]; (vi) 100 GHz Si$_3$N$_4$ microcomb[33]; (vii) 22 GHz fibre-stabilized SiO$_2$ microcomb[21]; (viii) MgF$_2$ microcomb[53]; (ix) 14 GHz MgF$_2$ microcomb pumped by an ultrastable laser[30]; and (x) 14 GHz microcomb-based transfer oscillator[31]. SSB, Single side band.

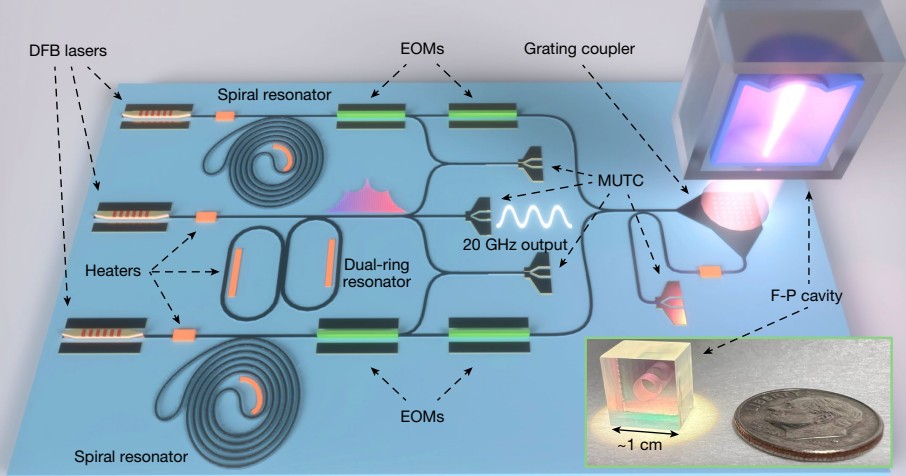

**Fig. 5 | Schematic design of a photonic microwave oscillator on a single chip.** The integrated system uses the same key photonic elements used in this work. Two spiral resonator SIL lasers are PDH locked to the same micro-F-P cavity with two EOMs in series for each SIL laser—the first for fast phase correction and the second for PDH side bands. The right side of the schematic shows the F-P cavity interface, where the two SIL laser paths are fed through an interferometer with an embedded polarization splitting grating. This serves as a reflection cancellation circuit while also shaping the planar waveguide mode to match the F-P mode[45]. The reflection from the F-P cavity is then detected by the right-most detector. The inset shows a photo of the miniature F-P cavity consisting of microfabricated mirrors[16], with overall volume of approximately 1 cm$^3$. Scale bar, approximately 1 cm. Illustration reproduced with permission from B. Long.

1,560 nm (ref. 6), spiral resonators[7] for SIL, a coupled-ring microcomb resonator[9], photodetectors[22] and a microfabricated F-P cavity that does not require high vacuum[16,18].

Previous work already laid out the steps for heterogeneous integration of lasers and Si$_3$N$_4$ resonators. For example, InP lasers and Si$_3$N$_4$ resonators have been integrated on the same chip with coupling between the optical gain and low-loss waveguide layers facilitated by adiabatic tapers, with resonator waveguide losses down to 0.5 dB m$^{-1}$ with a second deeply buried Si$_3$N$_4$ waveguide layer[15,41]. This same heterogeneous integration with ultrahigh-Q resonators promises isolator-free operation[41]. A similar strategy has been used for laser integration with 780-nm-thick Si$_3$N$_4$ anomalous dispersion microcombs on the same chip[42], which can be applied to the 100-nm-thick Si$_3$N$_4$ zero GVD microcombs used in this work. Furthermore, laser integration with modulators and detectors has also been previously demonstrated[43] and can be utilized for full integration of all the optical components comprising the PDH locking system[44].

The integration of the active and passive components on a single platform greatly reduces loss (between fibre and chip) and removes the need for the optical amplifiers we have used in the present work. In such a case, a few tens of milliwatts of optical power is required to pump the resonator such that a comb with a few milliwatts of optical power and several microwatts per mode is realistic. Additionally, for the SIL lasers, only several milliwatts of DFB optical power is required to provide a hundred microwatts of optical power to heterodyne with the comb and achieve the signal-to-noise ratio (SNR) necessary to match the performance presented in this work. For recent integrated lasers, these powers are realistic[15]. Additional considerations on required optical power are discussed in Methods.

Integration of the F-P cavity has been an outstanding challenge, but recent developments in microfabricated mirrors[16] and compact thermal-noise-limited F-P designs[17] provide new integration opportunities. Critically, it has been shown that 2P-OFD does not require F-P operation in high vacuum because of CMR[18], significantly simplifying future integration. Figure 5 shows a 1 cm$^3$ cavity with fabricated micromirrors and details on an integration strategy with the SIL lasers and microcomb. A planar waveguide feeds an inverse-designed polarization splitting grating embedded in an interferometer, which serves to shape the beam for coupling light to the cavity while also providing the cavity-reflected PDH locking signal and laser isolation[45]. Preliminary measurements described in Methods demonstrate the feasibility of this approach. The F-P cavity and a gradient-index (GRIN) lens can be bonded on top of the polarization splitting grating in a hybrid flip-chip fashion for a single-chip, cavity-integrated, low-noise microwave generator unit.

In the integration scheme, the acousto-optic modulators (AOMs) can be replaced with a combination of slow feedback to the integrated heaters or piezoelectric components[46] in the spirals and fast feedback to DFB current and EOMs[24]. The thermal tuning can reach a bandwidth of a few kilohertz[41,47], whereas the fast feedback with a bandwidth of several megahertz could be provided by the EOM or current modulation[43,44]. We estimate that this combination can provide 40 dB feedback gain at 10 kHz offset frequency to match the phase noise performance of the presented work. To further reduce the size of the entire system, the modulation frequencies for the PDH locking, as well as for phase locking of the intermediate frequency, could be synthesized by using a direct digital synthesizer, clocked by the microwave derived from the microcomb itself[48,49].

In summary, we have demonstrated an integrated photonic approach to OFD that produces 20 GHz microwave signals with phase noise of −135 dBc Hz$^{-1}$ at 10 kHz offset, a value typical of much larger existing commercial systems. This is accomplished with a unique combination of low-noise integrated lasers, an efficient dark soliton frequency comb and new advances in a miniature F-P optical cavity. Significantly, our approach provides a route to full integration on a single chip with volume of the photonic components on the order of 1 cm$^3$. This advance in integrated photonic low-noise microwave generation holds promise for compact, portable and low-cost microwave synthesis for a wide variety of demanding applications in navigation, communications and precise timing.

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

# Article

## Methods

### Characterization of the noise contributions to the microwave noise

Extended Data Fig. 1 shows the phase noise of the various components that contribute to the generated 20 GHz signal. This includes the in-loop phase noise and SNR of each continuous wave laser and the intermediate frequency, the SNR of the 20 GHz carrier and intensity noise projected onto the microwave phase because of finite amplitude-to-phase rejection. The in-loop phase noise of continuous wave lasers was estimated as described in ref. 54. The SNR of the continuous wave lasers is estimated in a similar way but when the lasers are unlocked and detuned from the F-P cavity resonance. The in-loop phase noise of the intermediate frequency was directly measured by using a commercial phase noise measurement system. Note that the aforementioned measured noise terms, excluding amplitude-to-phase conversion, were scaled down by 30 dB to account for their reduction because of OFD. The amplitude-to-phase conversion was determined from the measured relative intensity noise of the microcomb on the MUTC photodetector after scaling down by the amplitude-to-phase rejection (40 dB). The rejection value was measured by modulating the microcomb optical power and comparing the strength of amplitude modulation tone with the resulting phase modulation on the 20 GHz carrier.

At frequencies below 1 kHz, the phase noise of the 20 GHz signal is limited by the electronic noise of the PDH locks, which could be improved by increasing the 2P-OFD phase noise reduction (that is, by using a microcomb with broader bandwidth). At frequencies above several tens of kilohertz, the phase noise increases because of the limited gain and bandwidth of the microcomb feedback loop. The phase noise of the 20 GHz signal at frequencies above 1 MHz follows the phase noise of the free-running signal, which can be found in ref. 9. High-frequency noise could be decreased by using a more stable free-running microcomb state or by improving the feedback bandwidth. As can be seen, in the range between 1 and 40 kHz, the listed noise contributions do not affect the microwave generation, which decreases as $1/f$. A possible source of noise in this region is microcomb noise that is uncorrelated between modes separated by approximately 640 GHz. Another possible limitation that restricts the noise floor is the limited servo gain, which already provides more than 50 dB reduction in that region. Mitigating these constraints would allow for a further decrease of the noise in that region by up to 10 dB.

### Electronics for microcomb stabilization and phase noise measurement

Extended Data Fig. 2a shows the electronics used for microcomb stabilization. Both beat notes between continuous wave lasers and the microcomb teeth are filtered, amplified and mixed together to produce the intermediate frequency ($f_{IF}$) at 2.6 GHz. The IF frequency is further amplified and mixed with a reference oscillator to produce an error signal for microcomb stabilization. The error signal is fed to a servo, which provides feedback to the current supply of the seed laser. Note that the noise of the reference oscillator in this servo is divided down by the 2P-OFD, and in a future implementation, it could be obtained from the division of the 20 GHz signal itself.

Extended Data Fig. 2b illustrates the experimental setup for measuring phase noise of the 20 GHz signal. To measure the phase noise of the microcomb, we used the ultrastable microwave from a self-referenced Er:fibre frequency comb[2] as a reference oscillator. This Er:fibre comb was used only for the measurement purpose. The microcomb optical signal and the reference Er:fibre comb are detected by using two MUTC detectors. Then, the 20 GHz signal under test and the reference 20 GHz are filtered, amplified and split. The reference microwave signal is further amplified to saturate the mixers. Each arm of the microwave signal from the microcomb is mixed with the reference microwave to produce two signals at 47 MHz. Two arms are used for cross correlation to remove the additional noise from the microwave amplifiers in the reference branch. The cross correlation was realized with a commercial phase noise analyzer.

The phase noise of the reference 20 GHz is shown in Extended Data Fig. 2c. To measure the phase noise of the reference signal, we cross correlated it with two microwave oscillators for 3 h. The approximated measurement floor, shown in Extended Data Fig. 2c, represents the phase noise of the microwave oscillators and the cross correlation gain (30 dB) because of the finite number of averages.

### Regenerative divide by two

Extended Data Fig. 3a shows the scheme of the regenerative divide by two, which consists of a double-balanced mixer, 10 GHz amplifier, power splitter and phase shifter[55]. The phase shifter is used to control the phase delay inside the divider. The input 20 GHz is amplified up to approximately +13 dBm to saturate the local oscillator port of the double-balanced mixer. The IF port with 10 GHz signal is then amplified and split to provide the output signal with power of approximately +10 dBm. The output 10 GHz is measured using the same setup as shown in Extended Data Fig. 2b, whereas the reference signal at 10 GHz is provided from the same fibre frequency comb.

Extended Data Fig. 3b provides the schematic of the setup for measuring the phase noise of the divider. The input signal is split and fed to two separate dividers. The outputs from the dividers are cross correlated with the phase noise analyser. Because of the cross correlation, the correlated noise between both signals (that of the input signal) is averaged out, providing the phase noise of the divider itself. The phase noise of the divider at carrier input frequencies 16 and 18 GHz is presented in Extended Data Fig. 3c.

### Optical power requirements for the integrated system

Here, we consider the required optical power to achieve the performance presented in the paper. In the experimental demonstration described in the main text, the purpose of the optical amplifiers was to compensate the coupling losses between the SIL lasers and the chips and between the chips and the lens fibres. This ensured that all photodetected signals had the power needed to increase the SNR, limited by the thermal noise, to the required level. Extended Data Fig. 4a shows the calculated achievable SNR as a function of microcomb optical power when compared with the noise floor that consists of thermal and shot noise. Here, we assume detection with the MUTC having quantum efficiency of 0.5. To reach the same performance that we demonstrate in this present work, but without optical amplification, would require only a few hundred microwatts of comb optical power on the 20 GHz photodetector. Extended Data Fig. 4b shows the calculated SNR of the beat notes against the optical power of the SIL laser for three different comb tooth powers. As can be seen, to reach the performance presented in this work, it would be sufficient to combine tens of microwatts of SIL laser power with a few microwatts of comb tooth power. In our work, the heterodyne detection of the beat note with the SIL lasers used comb teeth with power of approximately 50 μW, whereas the total comb power before optical couplers was 60 mW. Thus, assuming that the losses after the microcomb generation are similar to this work, to provide comb teeth power of a few microwatts with a similar spectrum shown in Fig. 2, only several milliwatts of the total microcomb optical power would be required. To provide tens of microwatts from the SIL reference laser for heterodyning, the optical power on the order of a few hundred microwatts from the spiral resonator is required. This implies only a few milliwatts of power directly from the DFB laser.

Nonetheless, the single-chip integration would mitigate the coupling losses and increase the system efficiency. Additionally, removing AOMs would improve the power of the reference SIL lasers by approximately 3–5 dB, which is the insertion loss of the AOMs we used. In conclusion, these numbers are realistic to achieve with present integrated lasers[15].

## Cavity with microfabricated mirrors

In this work, we propose to use a cavity with microfabricated mirrors[16], which can be realized in a cube with overall volume below 1 cm$^3$ and does not require operation in vacuum (inset in Fig. 5). Extended Data Fig. 5a shows the ring-down measurement of such a cavity at 1,550 nm. The cavity has 60 kHz linewidth and provides finesse of 428,000 and $Q \approx 2$ billion. This measurement indicates that such a compact cavity has the required linewidth to provide the necessary SNR in the PDH locking.

Extended Data Fig. 5b schematically shows the cavity integration with the chip. The light from the waveguide is redirected to the integrated grating coupler. The output light from the grating is collected with a GRIN lens that optimally matches the output of the grating to the cavity mode. To avoid the back reflection from the cavity and reroute the reflected signal to another port for PDH detection, we design the on-chip circuits that are shown in Extended Data Fig. 5c and described in more details in ref. 45. The input light is split 50/50 and then recombined at the on-chip polarization splitting grating coupler[56] into a circularly polarized beam that is directed into free space. The reflected light from the cavity will switch the handedness and be collected by the polarization splitting grating coupler. By interfering two polarizations of reflected light in a 50/50 on-chip beam splitter, the back reflection is cancelled, and the signal rerouting is achieved.

To characterize the performance of this system, we use an SMF28 fibre to collect the output light from the polarization splitting grating coupler. The other end of fibre is connected to the GRIN lens, which then transfers the light into the cavity. The system was used as a proof of principle to measure the transmission and back reflection of the grating-GRIN-cavity system; those results are shown in Extended Data Fig. 5d. The insertion loss of the system is approximately −7 dB, whereas back reflections toward the laser source are reduced by −17 dB. Note that this system includes additional losses because of the inserted fibre, and the efficiency can be further improved by avoiding that step. Also, the back-reflection suppression ratio can be further improved by implementing an on-chip tunable coupler to achieve a more ideal 50/50 splitting ratio. These results provide confidence that the micro-fabricated cavity can be successfully integrated onto a planar chip.

## Data availability

All data for the figures in this manuscript are available at https://doi.org/10.6084/m9.figshare.24243511.

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

**Acknowledgements** We thank B. Long for the illustration in Fig. 5 and K. Chang and N. Hoghooghi for comments on the manuscript. Commercial equipment and trade names are identified for scientific clarity only and do not represent an endorsement by NIST. The research reported here performed by W.Z., V.I. and A.M. was carried out at the Jet Propulsion Laboratory at the California Institute of Technology under a contract with the National Aeronautics and Space Administration. This research was supported by the DARPA GRYPHON Program (grant HR0011-22-2-0009), the National Aeronautics and Space Administration (grant 80NM0018D0004) and NIST.

**Author contributions** P.T.R., J.E.B., K.J.V., A.M., F.Q. and S.A.D. conceived the experiment and supervised the project. I.K., W.G. and S.A.D. wrote the paper with input from all authors. I.K. and W.G. together with Q.-X.J. and J.G. built the experiment and performed the optical frequency division experiment. L.W. prepared the distributed feedback laser butterfly packages for the experiment. Q.-X.J., J.G., W.J., L.W., C.X. and L.C. prepared the microcomb and spiral resonators for the experiment. M.L.K. and F.Q. built the Fabry–Pérot cavity. D.L., T.N., C.A.M., Y. Liu and F.Q. provided the optically derived microwave reference and aided in the microwave phase noise measurement system. P.S., S. Hanifi and S.M.B. provided the regenerative divide-by-two circuit. H.C., N.J., S. Halladay, Z.D., Y. Luo, O.M., F.Q. and P.T.R. contributed to the cavity integration scheme. W.Z., V.I. and A.M., contributed to phase noise limitation analysis and system integration. J.B. and J.C.C. provided modified unitravelling carrier detectors. All authors contributed to the system design and discussion of the results.

**Competing interests** The authors declare no competing interests.

**Additional information**
**Correspondence and requests for materials** should be addressed to Igor Kudelin or Scott A. Diddams.

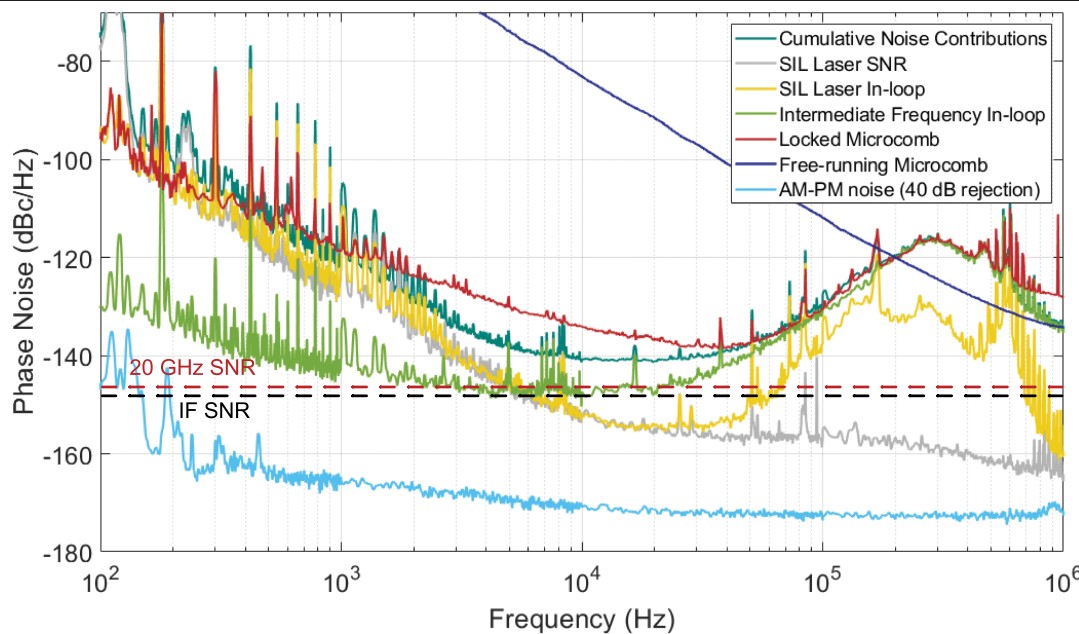

**Extended Data Fig. 1 | Noise contributions to the 20 GHz microwave.** Cumulative noise represents the quadrature sum of all the shown noise terms. In-loop laser noise is shown only for a single SIL laser, since the noise of both SIL lasers are the same. Red (black) dotted line shows SNR limits of the 20 GHz microwave (intermediate frequency).

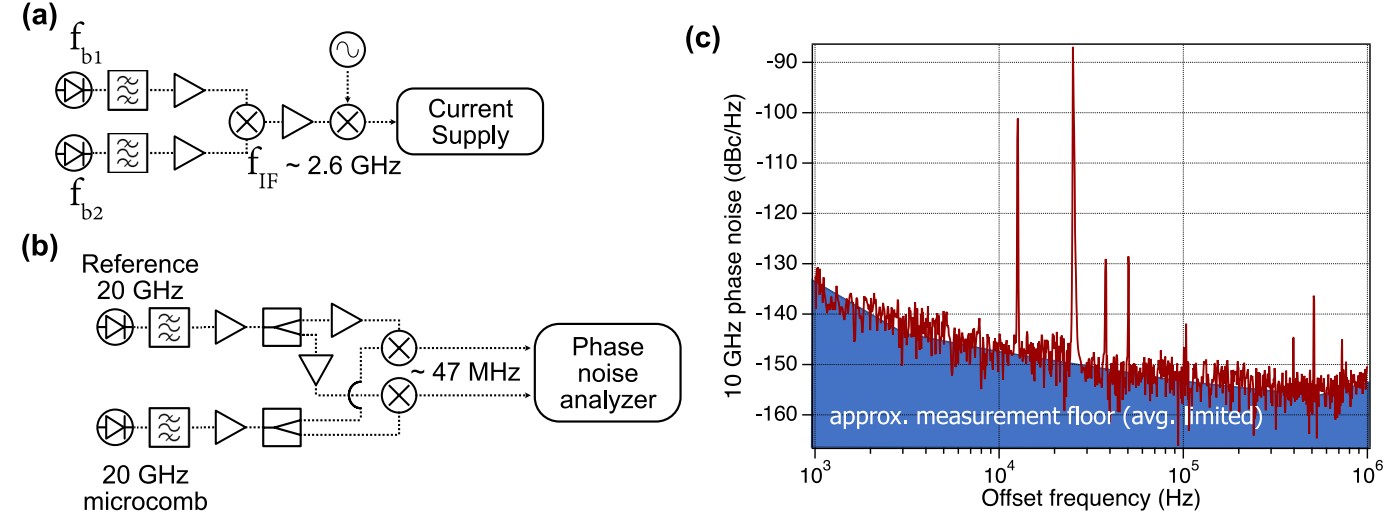

**Extended Data Fig. 2 | Microwave electronics used for stabilization and measurements. (a)** Electronics used to provide feedback for microcomb stabilization. **(b)** Phase noise measurement setup of 20 GHz microwave signal. **(c)** Phase noise of the reference 20 GHz signal used for cross correlation.

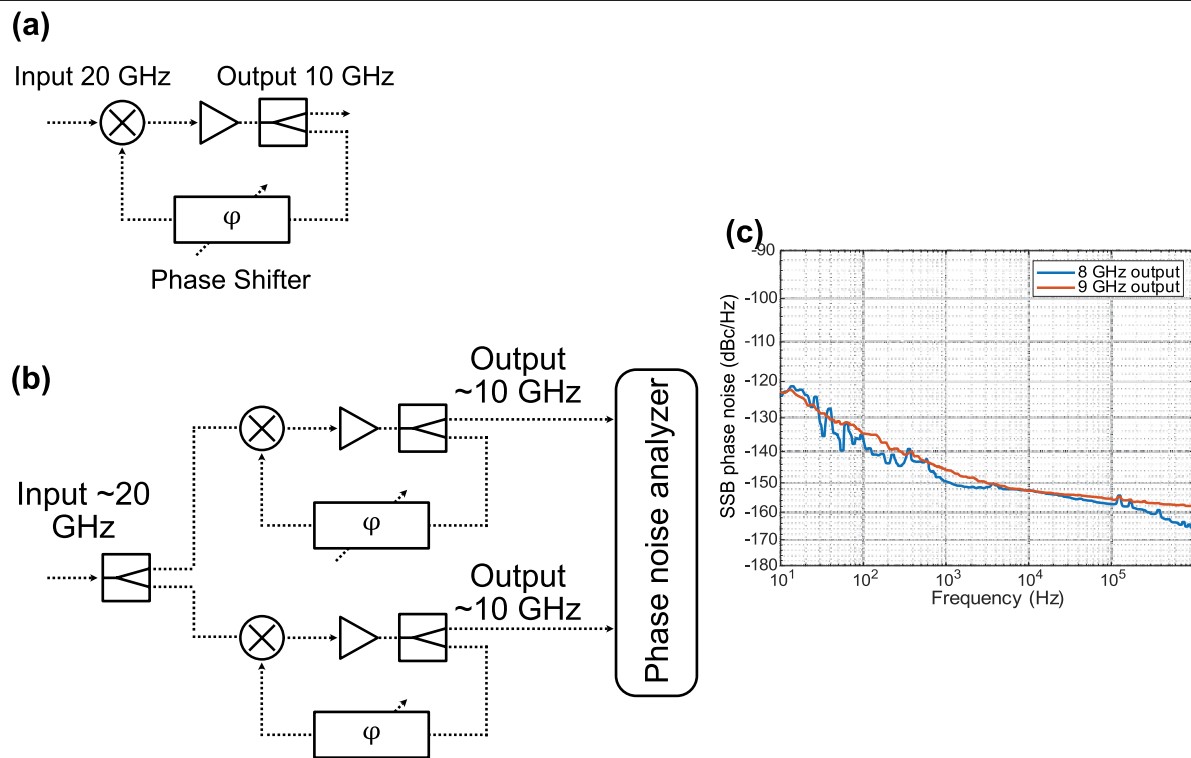

**Extended Data Fig. 3 | Regenerative divide-by-2. (a)** Setup of the regenerative frequency divide-by-2. **(b)** Experimental setup to measure the phase noise of the divider. **(c)** Phase noise of the regenerative divider referenced to the output carriers of 8 and 9 GHz, respectively.

**(a)**

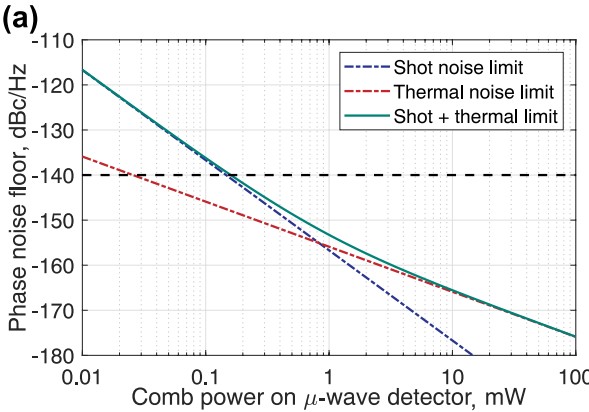

**(b)**

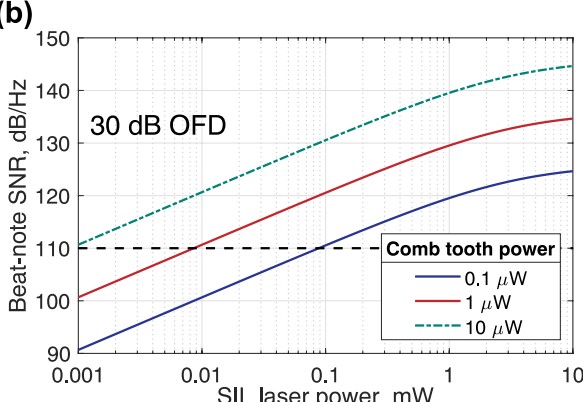

**Extended Data Fig. 4 | Optical power requirements.** Estimated optical power to reach a specified phase noise floor or SNR in **(a)** the detection of the 20 GHz microcomb, and **(b)** the beat note between the SIL lasers and microcomb. The dotted line represents the required performance to achieve the results presented in the paper. The calculations assume a photodetector quantum efficiency of 0.5.

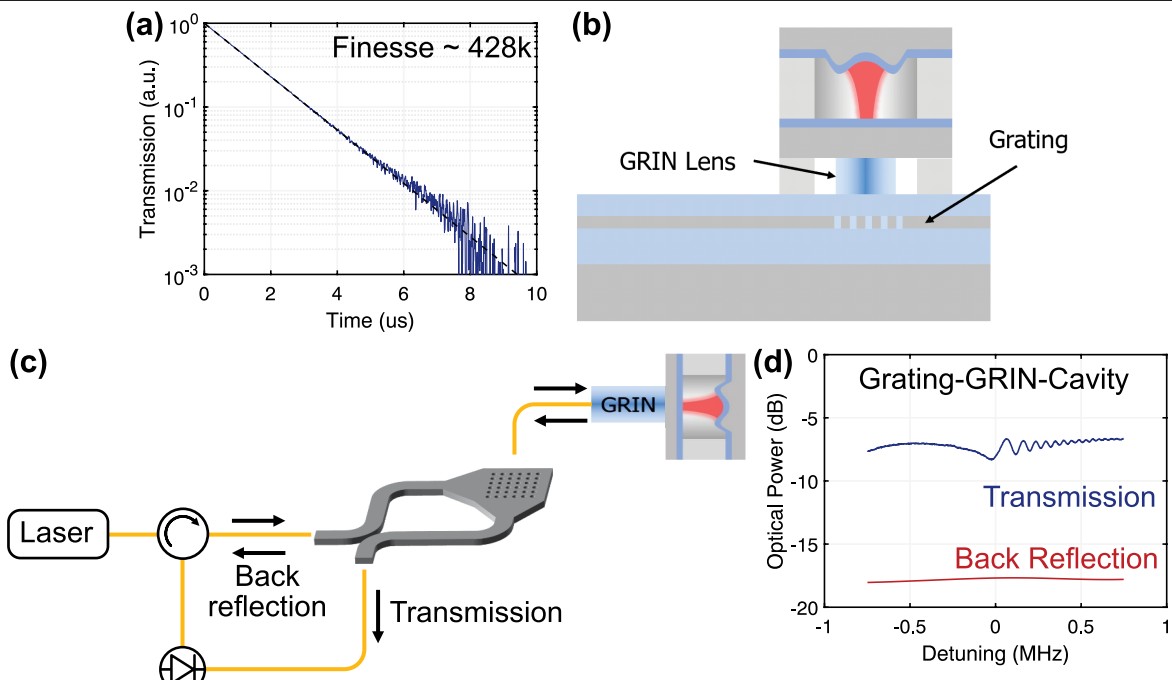

**Extended Data Fig. 5 | Integration and characterization of cavity with micro-fabricated mirrors. (a)** Transmission ringdown measurement of the 1 cm$^3$ cavity. **(b)** Scheme for coupling the light to the cavity through an integrated grating and GRIN lens. **(c)** Test setup for measuring the coupling to the micro F-P cavity. **(d)** Optical power of transmitted and back reflected light through Grating-Fibre-GRIN-Cavity system as a function of different input laser frequency.