## [Peer Review File · Nature]

Manuscript Title: Photonic chip-based low noise microwave oscillator

Reviewer Comments & Author Rebuttals

Reviewer Reports on the Initial Version:

Referees' comments:

Referee #1 (Remarks to the Author):

This manuscript discusses exciting new progress in integrated photonics based generation of low phase noise microwaves. By using an approach termed two point optical frequency division (2P-OFD) and a variety of state-of-the-art integrated photonic components, the group achieves phase noise on a 20 GHz carrier as low as -96 dBc/Hz (-135 dBc/Hz) at 100 Hz (10 KHz) frequency offset. These values already show improvement compared to standard electronic frequency generation equipment. Although significantly lower phase noise has been demonstrated in frequency division experiments based on mode-locked lasers, here the excitement has to do with the prospects to realize OFD based on highly compact integrated photonics components. This offers potential for synthesis of very low noise microwaves in a much smaller package and with greatly reduced power, which can impact a variety of applications related to timing, navigation, communications and spectroscopy.

This work employs an impressive roster of state-of-the-art integrated photonic components. The list includes:

- A pair of DFB lasers, each of which is self-injection locked to SiN spiral cavities with Q-factors above 10^8 .
- A novel integrable miniature Fabry-Perot (FP) cavity developed recently by a subset of the authors with finesse of $\sim 900,000$ and a Q-factor of $\sim 5 \times 10^9$ that does not require to operate in vacuum. Both of the DFB lasers are locked to difference resonances of the same FP cavity, which allows for large common-mode rejection of the relative phase between the two lasers. We are told the common-mode rejection can reach 40 dB for frequencies spaced by up to 1 THz. Together with the 2P-OFD, very large phase noise reduction is expected.
- A microresonator-based Kerr comb (microcomb) generated at 20 GHz repetition rate, using a dual coupled ring resonator fabricated using low confinement SiN waveguides from a CMOS foundry. The resonator is pumped by a DFB laser which is self-injection locked to the cavity, without optical amplification, generating a dark soliton comb of order 1 THz wide. Dark soliton microcombs generally have much higher power per line, and much higher pump power to comb conversion efficiency, compared to the more common soliton microcombs. This enhances prospects to realize a frequency division system with low power requirements. The comb is used to beat with both of the DFB lasers

which are spaced by ~ 640 GHz, allowing for 2P-OFD with a frequency division factor of 32. This results in reduction in the phase noise power spectrum by an additional factor of 32^2 (~ 30 dB).

- A MUTC photodiode (modified uni-traveling carrier photodiode) which provides high linearity (low amplitude-to-phase noise conversion) and is capable of high microwave output powers. Here a microwave power of -10 dBm is generated at 20 GHz with 5 mA average photocurrent.

Figure 4 shows a nice comparison of results from 10 different experiments pursuing low phase noise microwave generation using microcomb technology. Specifically Fig. 4 portrays these results along two axes: the reported phase noise at 10 KHz offset frequency and the authors' assessment of the level of integration present in the various experiments. The work in this manuscript has the lowest value of phase noise reported in any of the microcomb papers as well as the highest degree of integration of any of the optical frequency division based microcomb experiments. Compared to the one stand-alone microcomb result, the level of integration is similar, but the phase noise here about 40 dB better. In comparison to other OFD-based microcomb papers, the only one that is close is reported in a recent preprint (ref. 32, arXiv:2305.13575).

Despite the afore-mentioned impressive use of integrated photonic components, there will still be significant work to further reduce footprint and power usage. For example, the current experiment uses three optical fiber amplifiers (EDFAs) as well as isolators after the spiral cavity self-injection locked DFB lasers, both of which are difficult to integrate. However, the authors do give a robust discussion of their vision of the path forward towards further integration and power reduction. Certainly there are still challenges to overcome, but their discussion succeeds to paint a convincing picture of the potential to eventually achieve a deployable technology through heterogeneous integration.

Overall, I view this to be an outstanding candidate for publication in Nature.

I have just a couple of small questions and comments.

- It would be helpful to make an explicit comparison also with the phase noise levels of existing electronics approaches, such as those of commercial frequency synthesizers.
- Authors state " $n+m$ is the value of the optical frequency division" (page 4, second column, m and n refer to comb line numbers), but from Fig. 1b I would expect the frequency division factor to be $|m-n|$. Please check and clarify as appropriate.
- Although as stated the integration vision discussion is already robust and cites a number of important references, in some places it would be helpful to add a little further information on some of the approaches proposed. As one example, the modulators mentioned in the context of replacing the AOMs refer (ref. 24) to a work using a discrete fiber coupled electro-optic modulator, which is not an integrated component. What platforms do authors consider as promising for realizing this function in form compatible with heterogeneous integration?

Referee #2 (Remarks to the Author):

The authors combine a range of established techniques, including two-point optical frequency division, self-injection locked integrated lasers, miniature FP cavity, dark-soliton frequency comb, and high-power and high-linearity modified untraveling-carrier photodetectors, to demonstrate a low-noise microwave generator. The results are solid. I agree that the crucial photonic components used here hold promise for potential heterogeneous integration onto a single chip. However, the demonstrated system appears to be relatively bulky, and the noise performance does not yet match that of commercial products with standard packaging, such as X-LNO from QuantX, and hQp Photonic microwave oscillator. Additionally, the underlying idea is well-established within the field and the manuscript does not introduce a new or groundbreaking concept. Given these considerations, the novelty and significance of this work may be somewhat limited for publication in a prestigious journal like Nature.

Specific comments are below

- 1) The phase noise of the demonstrated microwave signal is noticeably suppressed at low frequency offsets within the locking bandwidth. This reduction is achieved through referencing to the miniature FP cavity. However, at higher offset frequencies, the dominant noise source stems from the dark-soliton frequency comb. It would be interesting to evaluate the noise performance across a wider frequency range to gain a comprehensive understanding of the system's behavior.
- 2) Is it possible to directly lock the two DFB lasers to the miniature FP cavity without involving Si₃N₄ resonators?
- 3) The demonstration employs two Si₃N₄ resonators to pre-stabilize the frequencies of two DFB lasers. Please comment on the feasibility of using just one Si₃N₄ resonator for this purpose? Will it lead to better noise suppression?
- 4) The demonstrated system relies on multiple microwave references for various functions, adding complexity. For instance, in the microcomb locking process on page 4, the authors phase-locked the intermediate frequency to a stable microwave reference. Is there a method to minimize this reliance on additional microwave references by adjusting the intermediate frequency closer to zero? If so, what would be the procedure for achieving this?

Referee #3 (Remarks to the Author):

In their manuscript on "Photonic chip-based low noise microwave oscillator" the authors devise and investigate a technology that allows very good phase noise. The scheme requires two injection locked diode lasers that are furthermore stabilized by being locked to a mini Fabry-Pérot cavity. Both of them are then used to stabilize a narrow optical frequency comb with the two point optical frequency division. The resulting phase noise stability is very impressive for an all compact and integratable device. It is still, however, short of the stability reached by fibre-optic/mode lock laser based OFD.

The work is original and the results significant and well reference, the conclusions are robust.

For the two reference lasers the authors mention that they are DFB lasers and for the pump laser of the soliton they mention a “commercial DFB” laser. Is there a difference, which type companies are they from?

The authors use a notch filter to filter out the pump laser, why not use a separate outcouple port from the dual ring such that only the in-coupled light is coupled out of the rings? The comb appears relatively flat, could the authors elaborate on this.

The subscript of the I_{IF} should I guess be IF on page 4 below second equation.

The authors use a self-referenced Er:fibre mode-locked laser, I guess this is just to measure their low phase noise and it is not necessary for the low phase noise of their OFD. If this is the case, please make it more clear, as such a system would neither be small nor cheap.

In the text “thick Si₃N₄” is used but not defined nor distinguished from their Si₃N₄ implementation.

Tiny suggestions:

Spelling of “combination” (page 6) and very many “this” in the last paragraph

Author Rebuttals to Initial Comments:

Referee #1

This manuscript discusses exciting new progress in integrated photonics based generation of low phase noise microwaves. By using an approach termed two point optical frequency division (2P-OFD) and a variety of state-of-the-art integrated photonic components, the group achieves phase noise on a 20 GHz carrier as low as -96 dBc/Hz (-135 dBc/Hz) at 100 Hz (10 KHz) frequency offset. These values already show improvement compared to standard electronic frequency generation equipment. Although significantly lower phase noise has been demonstrated in frequency division experiments based on mode-locked lasers, here the excitement has to do with the prospects to realize OFD based on highly compact integrated photonics components. This offers potential for synthesis of very low noise microwaves in a much smaller package and with greatly reduced power, which can impact a variety of applications related to timing, navigation, communications and spectroscopy.

This work employs an impressive roster of state-of-the-art integrated photonic components. The list includes:

A pair of DFB lasers, each of which is self-injection locked to SiN spiral cavities with Q-factors above 10^8 .

A novel integrable miniature Fabry-Perot (FP) cavity developed recently by a subset of the authors with finesse of $\sim 900,000$ and a Q-factor of $\sim 5 \times 10^9$ that does not require to operate in vacuum. Both of the DFB lasers are locked to difference resonances of the same FP cavity, which allows for large common-mode rejection of the relative phase between the two lasers. We are told the common-mode rejection can reach 40 dB for frequencies spaced by up to 1 THz. Together with the 2P-OFD, very large phase noise reduction is expected.

A microresonator-based Kerr comb (microcomb) generated at 20 GHz repetition rate, using a dual coupled ring resonator fabricated using low confinement SiN waveguides from a CMOS foundry. The resonator is pumped by a DFB laser which is self-injection locked to the cavity, without optical amplification, generating a dark soliton comb of order 1 THz wide. Dark soliton microcombs generally have much higher power per line, and much higher pump power to comb conversion efficiency, compared to the more common soliton microcombs. This enhances prospects to realize a frequency division system with low power requirements. The comb is used to beat with both of the DFB lasers which are spaced by ~ 640 GHz, allowing for 2P-OFD with a frequency division factor of 32. This results in reduction in the phase noise power spectrum by an additional factor of 32^2 (~ 30 dB).

A MUTC photodiode (modified uni-traveling carrier photodiode) which provides high linearity (low amplitude-to-phase noise conversion) and is capable of high microwave output powers. Here a microwave power of -10 dBm is generated at 20 GHz with 5 mA average photocurrent.

Figure 4 shows a nice comparison of results from 10 different experiments pursuing low phase noise microwave generation using microcomb technology. Specifically Fig. 4 portrays these results along two axes: the reported phase noise at 10 KHz offset frequency and the authors' assessment of the level of integration present in the various experiments. The work in this manuscript has the lowest value of phase noise reported in any of the microcomb papers as well as the highest degree of integration of any of the optical frequency division based microcomb experiments. Compared to the one stand-alone microcomb

result, the level of integration is similar, but the phase noise here about 40 dB better. In comparison to other OFD-based microcomb papers, the only one that is close is reported in a recent preprint (ref. 32, arXiv:2305.13575).

Despite the afore-mentioned impressive use of integrated photonic components, there will still be significant work to further reduce footprint and power usage. For example, the current experiment uses three optical fiber amplifiers (EDFAs) as well as isolators after the spiral cavity self-injection locked DFB lasers, both of which are difficult to integrate. However, the authors do give a robust discussion of their vision of the path forward towards further integration and power reduction. Certainly there are still challenges to overcome, but their discussion succeeds to paint a convincing picture of the potential to eventually achieve a deployable technology through heterogeneous integration.

Overall, I view this to be an outstanding candidate for publication in Nature.

Answer: We appreciate the positive endorsement and comments provided by the reviewer.

I have just a couple of small questions and comments.

Q1. It would be helpful to make an explicit comparison also with the phase noise levels of existing electronics approaches, such as those of commercial frequency synthesizers.

Answer: We agree with the comment and have now included commercial frequency synthesizers in the comparison plot (Fig 4).

Q2. Authors state “ $n+m$ is the value of the optical frequency division” (page 4, second column, m and n refer to comb line numbers), but from Fig. 1b I would expect the frequency division factor to be $|m-n|$. Please check and clarify as appropriate.

Answer: We thank the Reviewer for pointing this out. The division factor of $(m-n)$ would be appropriate if the m^{th} and n^{th} modes are numbered from the zero frequency. However, in the manuscript we numbered the modes with integers beginning with mode zero being the central (seed) frequency ν_0 . This also follows from the equation for the beat-notes, where the beat frequencies with the comb lines are: $\nu_0 - n \cdot f_{\text{rep}}$ and $\nu_0 + m \cdot f_{\text{rep}}$. In this case, the division factor is $(m+n)$. To avoid potential confusion, we now explicitly mention in the text that the numbering is from the central frequency (page 4 right column).

Q3. Although as stated the integration vision discussion is already robust and cites a number of important references, in some places it would be helpful to add a little further information on some of the approaches proposed. As one example, the modulators mentioned in the context of replacing the AOMs refer (ref. 24) to a work using a discrete fiber coupled electro-optic modulator, which is not an integrated component. What platforms do authors consider as promising for realizing this function in form compatible with heterogeneous integration?

Answer: The integration of the electro-optic modulator with appropriate references is mentioned in the manuscript earlier on page 5 (lower part of the right column) as: “Laser integration with modulators, detectors, and optical amplifiers based on the heterogeneous InP/Si platform has also been previously demonstrated [48, 49] and can be utilized for full integration of the optical components comprising the PDH locking system [50].” Thus, we propose a heterogeneous InP/Si/SiN platform for the realization of the single-chip microwave source. Reference 24 (which is now ref 25) was provided to indicate that by providing the error signal to a phase modulator it is possible to correct high-frequency phase fluctuations of the CW lasers for efficient PDH locking. However, we adjusted the references in this paragraph to make it clear that the EOM can be used to correct the phase fluctuation, and that integrated EOMs are available.

Referee #2

The authors combine a range of established techniques, including two-point optical frequency division, self-injection locked integrated lasers, miniature FP cavity, dark-soliton frequency comb, and high-power and high-linearity modified untraveling-carrier photodetectors, to demonstrate a low-noise microwave generator. The results are solid. I agree that the crucial photonic components used here hold promise for potential heterogeneous integration onto a single chip. However, the demonstrated system appears to be relatively bulky, and the noise performance does not yet match that of commercial products with standard packaging, such as X-LNO from QuantX, and hQp Photonic microwave oscillator. Additionally, the underlying idea is well-established within the field and the manuscript does not introduce a new or groundbreaking concept. Given these considerations, the novelty and significance of this work may be somewhat limited for publication in a prestigious journal like Nature.

Answer: We appreciate the Reviewer's comment but want to emphasize that the significance of the architecture and combination of core components described in the submitted manuscript goes beyond the size of the auxiliary elements of the demonstration system. Our work represents the first time that integrated photonic elements, consisting of self-injection locked lasers, a dual ring microresonator comb, a miniature reference cavity, and high-linearity photodiodes **have been used together** in an integrable architecture that generates microwave signals with significantly low phase noise. This is a nontrivial result that required advances in the device physics and the design and operation of the integrated photonics, together with an engineered balance of performance versus ultimate size, power and volume.

Beyond the significant demonstration of low-noise microwaves with integrated photonics, in the manuscript we map a clear and realistic approach to achieve the goal of full integration; that includes power considerations to avoid the use of EDFAs, integrated PDH locking, and the further miniaturization and integration of the reference cavity. We would not expect that our system could reach the performance of oscillators made by QuantX or hQphotonics, as those involve key components with liter-sized volume with limited options for further size reduction. On the other hand, we show how our entire system can be implemented in a much smaller size of 10^{-2} liters with advanced fabrication that will be important for widespread applications beyond the laboratory environment. Moreover, the phase noise of our system can be further improved by exploiting a higher OFD value or by reducing the noise in the servo-loops to achieve higher common mode rejection. As such, we affirm that the integration of leading technologies to achieve low-SWaP microwave generation is significant on its own and provides a valuable contribution to the field.

Specific comments are below

Q1. *The phase noise of the demonstrated microwave signal is noticeably suppressed at low frequency offsets within the locking bandwidth. This reduction is achieved through referencing to the miniature FP cavity. However, at higher offset frequencies, the dominant noise source stems from the dark-soliton frequency comb. It would be interesting to evaluate the noise performance across a wider frequency range to gain a comprehensive understanding of the system's behavior.*

Answer: The observation of the reviewer is correct. At the lower offset frequencies, the phase noise is limited by noise from the PDH locking. At higher frequencies (above the feedback bandwidth of ~ 300 kHz),

the phase noise tends to follow the noise of the free running microcomb, reaching the SNR limited noise floor of -145 dBc/Hz at offset frequencies close to 10 MHz. The phase noise performance at higher offset frequencies of the free-running microcomb can be found in Ref. 21 [Q.-X. Ji, et al., "Engineered zero-dispersion microcombs using cmos-ready photonics," *Optica*, vol. 10, no. 2, pp. 279–285, 2023]. We also included this statement in the Supplementary Information.

Q2. *Is it possible to directly lock the two DFB lasers to the miniature FP cavity without involving Si3N4 resonators?*

Answer: Yes, the DFB lasers could be directly locked to the FP cavity, and a subset of the authors have tried a version of this already. However, free-running DFB lasers exhibit high frequency noise, making it nearly impossible to reach the thermal noise floor of the FP cavity without more gain in the PDH feedback loop. That problem would be particularly severe considering the fully-integrated design where the feedback gain is limited by the applied power to the EOM. Moreover, even with more gain, the phase noise at higher offset frequencies (at and beyond the servo bandwidth) would be significantly higher compared to the results we present. An attempt to PDH lock an integrated laser without pre-stabilization is provided in the following reference [Liron Stern, et al. "Ultra-precise optical-frequency stabilization with heterogeneous III–V/Si lasers," *Opt. Lett.* **45**, 5275-5278 (2020)]. Although the Authors of this paper used a laser with better noise performance than that of a DFB, they did not reach the thermal noise limit of their reference, which has a still higher thermal noise floor compared to the Fabry-Perot cavity used in our work. This highlights the importance of pre-stabilization of DFB lasers, that people addressed before, for example, by using fiber optics [Fordell, Thomas, et al. "Pre-stabilization of a distributed feedback diode laser for locking to a high-finesse cavity." *2016 Conference on Precision Electromagnetic Measurements (CPEM 2016)*. IEEE, 2016]. Similarly, passive pre-stabilization through self-injection locking to the Si3N4 spiral resonator is a very efficient approach to significantly reduce the phase noise of the entire system, but in a smaller and integrated volume. On page 3 we now include an additional sentence on the importance of the pre-stabilization of the DFB lasers in the main text.

Q3. *The demonstration employs two Si3N4 resonators to pre-stabilize the frequencies of two DFB lasers. Please comment on the feasibility of using just one Si3N4 resonator for this purpose? Will it lead to better noise suppression?*

Answer: We appreciate the Reviewer's suggestion for simplification of the setup. This is something we have considered; however, it would not lead to better performance. In the case of a single Si3N4 spiral resonator, the absolute phase noise of each DFB laser would still follow the thermal-noise limit of the Si3N4 spiral resonator. While the relative stability between two SIL DFB lasers would be improved due to common mode rejection, it would not matter after PDH locking to the cavity, when the final relative stability of the DFB lasers is limited by the servo loop noise. Note that the common mode rejection in the Si3N4 spiral resonator is different from the common mode rejection in FP cavity as discussed in this reference: Y. Liu, D. Lee, N. Jin, C. A. McLemore, Y. Luo, M. Kelleher, P. Rakich, S. A. Diddams, and F. Quinlan, "High Finesse, Air-Gap Optical Reference Cavity for Low Noise Microwave Generation," in *CLEO 2023, Technical Digest Series* (Optica Publishing Group, 2023), paper SM2K.4. Additionally, using a single Si3N4 spiral resonator would require two integrated WDMs (to combine the inputs and separate the output), leading to extra losses.

Q4. *The demonstrated system relies on multiple microwave references for various functions, adding complexity. For instance, in the microcomb locking process on page 4, the authors phase-locked the intermediate frequency to a stable microwave reference. Is there a method to minimize this reliance*

on additional microwave references by adjusting the intermediate frequency closer to zero? If so, what would be the procedure for achieving this?

Answer: We thank the Reviewer for this comment. There is a solution that we have demonstrated previously that can be implemented to remove the need for additional microwave references. This would involve using the repetition rate of the microcomb as the reference clock for all auxiliary frequencies in the system. For example, the detected microcomb repetition rate can be frequency divided and used as the clock for a Direct Digital Synthesizer (DDS), that subsequently synthesizes any frequencies needed for the PDH system or auxiliary phase locks. We have demonstrated the core elements of such an approach in earlier papers. See for example:

- Diddams, Scott A., et al. "An optical clock based on a single trapped 199Hg^+ ion." *Science* 293.5531 (2001): 825-828.
- Fortier, T. M., et al. "Optically referenced broadband electronic synthesizer with 15 digits of resolution." *Laser & Photonics Reviews* 10.5 (2016): 780-790.
- Kudelin, Igor, et al. "Tunable opto-electronic synthesizer at 10 GHz with ultralow phase noise." *2023 Conference on Lasers and Electro-Optics (CLEO)*. IEEE, 2023.

Slow changes of the comb repetition rate could ultimately be servo controlled to a GPS-derived signal via temperature control of the microcomb itself.

Beyond this, we note that there are multiple ways in which we can physically adjust and align the various beat frequencies, including the IF frequency. Firstly, depending on the location of the reference CW frequencies relative to the comb lines, the intermediate frequency can be derived as a sum or difference between the beat-notes. Thus, the IF can be reduced by locking the reference CW lasers to different modes of the FP cavity. Additionally, the beat-notes can be adjusted by changing the wavelengths of the seed DFB laser for microcomb generation. Lastly, the IF frequency can be tuned by changing the repetition rate of the microcomb by controlling the temperature of the chip. Note that the changed of the IF will be magnified by $(n+m)$ compared to the change of the repetition rate. By using these degrees of freedom, it is possible to reduce the IF.

We have added brief statements on the last page in the text to clarify these points.

Referee #3

In their manuscript on “Photonic chip-based low noise microwave oscillator” the authors devise and investigate a technology that allows very good phase noise. The scheme requires two injection locked diode lasers that are furthermore stabilized by being locked to a mini Fabry-Pérot cavity. Both of them are then used to stabilize a narrow optical frequency comb with the two point optical frequency division. The resulting phase noise stability is very impressive for an all compact and integratable device. It is still, however, short of the stability reached by fibre-optic/mode lock laser based OFD.

The work is original and the results significant and well reference, the conclusions are robust.

Answer: We thank the Reviewer for the positive appreciation of the provided work.

Q1. *For the two reference lasers the authors mention that they are DFB lasers and for the pump laser of the soliton they mention a “commercial DFB” laser. Is there a difference, which type companies are they from?*

Answer: All three used DFB lasers are commercial and were provided by PhotonX and EMCORE Corporation. There is no difference between them except the chosen operation wavelengths. In the manuscript, we added “commercial” to each DFB used, so there would be no confusion.

Q2. *The authors use a notch filter to filter out the pump laser, why not use a separate outcouple port from the dual ring such that only the in-coupled light is coupled out of the rings? The comb appears relatively flat, could the authors elaborate on this.*

Answer: We completely agree with the Reviewer on this point. The drop-port could be used to suppress the central frequency and to flatten the comb. However, the chip we used had a damaged drop-port with increased losses. We now include the possibility of using the drop-port in the text.

Q3. *The subscript of the I_{IF} should I guess be IF on page 4 below second equation.*

Answer: We thank the Reviewer for noticing this. We have corrected our mistake.

Q4. *The authors use a self-referenced Er:fibre mode-locked laser, I guess this is just to measure their low phase noise and it is not necessary for the low phase noise of their OFD. If this is the case, please make it more clear, as such a system would neither be small nor cheap.*

Answer: Yes, this is the case. We amended the description of the measurement to make this more clear and have moved this text to the Supplement to save space. The new text in the Supplement now reads: "To measure the phase noise of the microcomb we used the ultra-stable microwave from a self-referenced Er: fiber frequency comb ~\cite{nakamura2020coherent} as a reference oscillator. This Er: fiber comb was used only for the measurement purpose."

Q5. *In the text "thick Si₃N₄" is used but not defined nor distinguished from their Si₃N₄ implementation.*

Answer: We thank the reviewer for pointing out this oversight. We have amended the text as the following: "A similar strategy has been employed for laser integration with 780-nm thick Si₃N₄, anomalous dispersion microcombs on the same chip ~\cite{xiang2021laser}, which can be applied to the 100-nm thick Si₃N₄, zero-GVD microcombs used in this work."

Q6. *Tiny suggestions: Spelling of "combination" (page 6) and very many "this" in the last paragraph*

Answer: We appreciate the attentive review of our manuscript. We have fixed the spelling and reduced the presence of "this" in the last paragraph.

Reviewer Reports on the First Revision:

Referees' comments:

Referee #1 (Remarks to the Author):

I am supportive of publication of the revised manuscript.

Referee #2 (Remarks to the Author):

I appreciate the authors for well addressing my comments, and I recommend the publication of their work.

Referee #3 (Remarks to the Author):

The authors did a fine job ob responding to the comments raised by myself and the other referees. The manuscript is now much improved and I have no further comments.

Author Rebuttals to First Revision:

We want to express our gratitude to all the referees for their constructive comments that improved our manuscript.

Referee #1

I am supportive of publication of the revised manuscript.

Referee #2

I appreciate the authors for well addressing my comments, and I recommend the publication of their work.

Referee #3

The authors did a fine job of responding to the comments raised by myself and the other referees. The manuscript is now much improved and I have no further comments.